# *Epsilon*-Caprolactam- and Nylon Oligomer-Degrading Bacterium *Brevibacterium epidermidis* BS3: Characterization and Potential Use in Bioremediation

**DOI:** 10.3390/microorganisms11020373

**Published:** 2023-02-01

**Authors:** Tatiana Z. Esikova, Ekaterina V. Akatova, Inna P. Solyanikova

**Affiliations:** 1Laboratory of Plasmid Biology, G.K. Skryabin Institute of Biochemistry and Physiology of Microorganisms, Pushchino Center for Biological Research of the Russian Academy of Sciences, prosp. Nauki 5, 142290 Pushchino, Russia; 2Laboratory of Ecological and Medical Biotechnology, Tula State University, Friedrich Engels Street 157, 300012 Tula, Russia; 3Laboratory of Microbial Enzymology, G.K. Skryabin Institute of Biochemistry and Physiology of Microorganisms, Pushchino Center for Biological Research of the Russian Academy of Sciences, prosp. Nauki 5, 142290 Pushchino, Russia; 4Regional Microbiological Center, Institute of Pharmacy, Chemistry and Biology, Belgorod National Research University, 308015 Belgorod, Russia

**Keywords:** biodegradation, bioremediation, *Brevibacterium*, *epsilon*-caprolactam, nylon-6 oligomers, 6-aminohexanoic acid, adipic acid

## Abstract

*epsilon*-Caprolactam (Caprolactam, CAP), a monomer of the synthetic non-degradable polymer nylon-6, is the major wastewater component in the production of caprolactam and nylon-6. Biological treatment of CAP, using microbes could be a potent alternative to the current waste utilization techniques. This work focuses on the characterization and potential use of caprolactam-degrading bacterial strain BS3 isolated from soils polluted by CAP production wastes. The strain was identified as *Brevibacterium epidermidis* based on the studies of its morphological, physiological, and biochemical properties and 16S rRNA gene sequence analysis. This study is the first to report the ability of *Brevibacterium* to utilize CAP. Strain BS3 is an alcalo- and halotolerant organism, that grows within a broad range of CAP concentrations, from 0.5 up to 22.0 g/L, optimally at 1.0–2.0 g/L. A caprolactam biodegradation experiment using gas chromatography showed BS3 to degrade 1.0 g/L CAP over 160 h. In contrast to earlier characterized narrow-specific CAP-degrading bacteria, strain BS3 is also capable of utilizing linear nylon oligomers (oligomers of 6-aminohexanoic acid), CAP polymerization by-products, as sole sources of carbon and energy. The broad range of utilized toxic pollutants, the tolerance for high CAP concentrations, as well as the physiological properties of *B. epidermidis* BS3, determine the prospects of its use for the biological cleanup of CAP and nylon-6 production wastes that contain CAP, 6-aminohexanoic acid, and low molecular weight oligomer fractions.

## 1. Introduction

*Epsilon*-Caprolactam (Caprolactam, CAP, the cyclic amide of 6-aminohexanoic acid) is one of the chemical products that are in high demand on the world market. CAP is used as a raw material to produce polycaprolactam, more known as nylon-6. This polymer is produced by the ring-opening polymerization of caprolactam, in which the monomeric unit, 6-aminohexanoic acid (6-AHA), is linked by amide bonds with a degree of polymerization > 100 [1]. The polymerization products contain the linear and cyclic oligomers of 6-aminohexanoic acid traditionally called nylon oligomers (OLN). Caprolactam, 6-aminohexanoic acid, and nylon oligomers are the major components of wastewater and solid wastes formed in the production of CAP and nylon-6 [2]. At present, the wastewater is incinerated, and the solid wastes are buried, which leads to the polluting soils and ground waters by caprolactam and its by-products. In several works, CAP is shown to cause dermatitis, fever, and seizures in humans, to induce DNA damage and cause chromosomal aberrations in mammals [3,4,5], as well as to inhibit the growth of microorganisms that play important roles in maintaining the ecological equilibrium of soils [6]. Due to its toxic effect on living organisms, this pollutant poses a hazard to the environment and human health.

To date, CAP-degrading bacteria from various taxonomic groups have been described: proteobacteria, actinobacteria, and spore-forming bacteria [7,8,9]. Some of them utilize the xenobiotic as a sole source of carbon and nitrogen; others require additional sources of carbon or nitrogen.

Although the linear nylon oligomers have structures like those of linear peptides found in biological systems and thus could probably be acted upon by proteases/peptidases/amidohydrolases produced by microorganisms, they are harder to degrade by microbes than CAP. Several bacterial strains have been described that are capable of transforming OLN to 6-aminohexanoic acid [2] and, in some cases, of utilizing them completely [1,10]. Earlier, we isolated and characterized CAP-degrading bacteria attributed to the genus *Pseudomonas* that do not degrade the linear OLN but transform them to corresponding dicarboxylic acids [11]. The diversity of microorganisms oxidizing nylon oligomers, and peculiarities of the biodegradation of these man-made compounds by them remain little investigated. The absence of any systemic research into microbial degraders of CAP and OLN hinders the development of safe and economical technologies for the biological cleanup of their production wastes and the bioremediation of polluted sites.

It is thought that the biodegradation of CAP and OLN in bacteria proceeds via the common biochemical pathway [2,11]. This suggests that caprolactam-degrading strains could use OLN as growth substrates. Still, the possibility of degrading CAP and OLN as sole sources of carbon without any additional growth factors by one bacterial culture has not been described. This work presents the results of investigating the morphological, physiological, and biochemical properties and the taxonomic status of the strain BS3, which possesses a unique capability of utilizing not only caprolactam but also the low molecular weight linear nylon oligomers (as exemplified by the 6-AHA dimer and trimer) as sole sources of carbon and energy. We also aim to assess the degradation potential of this strain and the prospects of its application for the biological cleanup of industrial wastes from the production of caprolactam and CAP-based polymers.

## 2. Material and Methods

### 2.1. Chemicals

All reagents (Sigma, St.Louis, MO, USA; Merck, Darmstadt, Germany; Amresco, Solon, OH, USA; Honeywell Fluka, Seelze, Germany) were of analytical grade. The linear dimer and trimer of 6-aminohexanoic acid were synthesized according to Kinoshita et al. [12]. The oligomers were assayed by thin-layer chromatography and mass spectrometry.

### 2.2. Bacterial Strain and Growth Conditions

Bacterial strain BS3 (VKM Ac-2833) has been isolated earlier from soil sampled in a CAP production territory (Shchekino, Tula Region, Russia) [13]. The bacterium was cultivated in a mineral salt medium (MSM) containing (g/L): KH_2_PO_4_, 0.2; K_2_HPO_4_, 0.6; NaCl, 0.3; NH_4_Cl, 0.4; CaCl_2_ × 2H_2_O, 0.1; MgSO_4_ × 7H_2_O, 0.2. The pH of the medium was adjusted to 7.5; then the medium was sterilized by autoclaving at 121 °C for 20 min. A sterile solution of trace elements in the amount of 1.0 mL containing (g/L): H_3_BO_3_, 0.5; CuSO_4_ × 5H_2_O, 0.04; FeCl_3_, 0.2; MnSO_4_ × H2O, 0.4; (NH_4_)_6_Mo_7_O_24_ × 4H_2_O, 0.2; ZnSO_4_ × 7H_2_O, 0.4, was introduced into the medium. CAP, 6-AHA, adipinic acid (ADA), and the linear oligomers of 6-AHA (dimer and trimer) at concentrations of 1.0 g/L (*w*/*v*) were used as sole sources of carbon and energy. The culture was grown at 28 °C on a shaker (180 rpm) or in Petri dishes containing an agarized medium. For the preparation of agar plates, the medium was supplemented with 2% agar.

The physiological characteristics and optimum growth conditions were studied using a tryptone-soy medium (TSM) of the following composition (g/L): soy extract, 30.0; casein hydrolysate, 5.0; yeast extract, 1.0; aminopeptide, 60 mL L^−1^; pH 7.5.

### 2.3. Microscopy

Light microscopy of samples in the phase contrast mode was carried out using a Nikon Eclipse Ci microscope (Nikon, Minato, Japan) equipped with a ProgRes SpeedXT camera (Jenoptic, Jena, Germany). Electron microscopy of thin sections was carried out according to the method described in [14]. To prepare ultrathin sections, double fixation with glutaraldehyde and osmium tetraxide was used followed by embedding the cells in an epoxy resin Epon 812. Sections were viewed through an electron microscope JEM-1400 (JEOL, Akishima, Japan) at an accelerating voltage of 80 kV.

### 2.4. Characterization of the Physiological and Biochemical Properties

Optimal growth conditions were investigated in a liquid TSM. Growth was examined at different temperatures (4, 10, 18, 25, 30, 37, 40, 42, and 45 °C) and pH values (from 4.0 up to 11.0, in increments of 0.5 pH unit) adjusted with appropriate buffers [15] and various NaCl concentrations (0, 0.5, 1.0, 2.5, 5.0, 7.5, 10.0, 12.5, 15.0, 17.5, 20.0, 25.0, *w*/*v*). Culture growth was monitored at OD590 using a UV Specord spectrophotometer (Carl Zeiss, Oberkochen, Germany). Each variant of the experiment was performed in triplicate; the mean values were used for the analysis.

The catalase activity was investigated by assessing bubble production in a 3% (*v*/*v*) hydrogen peroxide solution. The oxidase activity was determined by the oxidation of tetramethyl-*p*-phenylenediamine [16].

Utilization of various substrates, enzyme activities, and other biochemical properties were tested by using Api 20E, Api 20NE, and Api 50CH strips (BioMerieux, Marcy-l’Étoile, France) according to the manufacturer’s instructions.

### 2.5. Identification of the Strain BS3

#### 2.5.1. 16. S rRNA Gene Sequencing and Phylogenetic Analysis

Genomic DNA was extracted with a ZR Fungal/Bacterial DNA MiniPrepTM Kit (Zymo Research, Irvine, CA, USA) in accordance with the manufacturer’s recommendations. Amplification of the 16S rRNA gene was performed with the GeneAmp PCR System 9700 with prokaryotic universal 16S rRNA gene primers: 27f (5′-AGAGTTTGATCCTGGCTCAG-3′) and 1492r (50-TACGGYTACCTTGTTACGACTT-30) [17]. The reaction products were separated by electrophoresis in 1% agarose gel. DNA fragments were isolated from agarose and purified using a ZymoClean Gel DNA Recovery Kit (Zymo Research, USA). Purified PCR products were sequenced on an ABI Prism 373 3130XL Genetic Analyzer (PerkinElmer, Waltham, MA, USA). The sequence obtained was compared with the sequences in GenBank databases using the BLAST software package [18]. Multiple alignments with sequences of the most closely related bacterial strains and calculations of levels of sequence similarity were carried out using the CLUSTAL X program [19].

The 16S rRNA nucleotide sequence of strain BS3 was deposited in the GenBank under accession number JN787123.

#### 2.5.2. G + C Analysis and DNA-DNA Hybridization

The G + C content of the genomic DNA for strain BS3 was determined by the thermal denaturation method using the DNA of *Escherichia coli* K12 as a standard [20]. DNA-DNA relatedness was studied using the fluorometric micro-well method [21]. Hybridization was performed in five replicates. The highest and lowest values obtained in each sample were excluded, and the means of the remaining three values were quoted as DNA-DNA relatedness values.

### 2.6. Caprolactam Degradation Experiment

Degradation of caprolactam was carried out by growing the bacterial strain in an MSM (pH 7.5) supplemented with 1.0 g/L substrate. The culture grown on the same medium up to the mid-exponential phase (OD590, 1.0) was used as inoculum. Cells were sedimented by centrifugation (5000 g, 10 min) and poured into flasks up to an OD590 of 0.05. Cultivation was done in 750 mL flasks with 100 mL of the medium for 160 h at 28 °C and 180 rpm. Samples to measure the optical density and the amount of CAP were taken every 20 h. Bacterial growth was assessed by the change of optical density, and the amount of substrate in samples (including control experiments without inoculation) was determined using gas chromatography.

### 2.7. Caprolactam Tolerance Test

Growth of BS3 at different CAP concentrations was examined in an MSM-containing substrate at 0.4, 0.5, 1.0, 2.0, 5.0, 10.0, 15.0, 20.0, 22.0, or 23.0 g/L. The conditions for the preparation of inoculum and cultivation of cells were the same as in the Caprolactam Degradation Experiment. Bacterial growth was assessed by measuring the optical density at 590 nm. The OD value was converted to dry cell weight (DCW) using a calibration curve. The duration of the lag period and the maximal specific growth rate (µmax) were calculated from the growth curves plotted for each concentration of substrate using Microsoft Office Excel 2007 [22].

### 2.8. Gas Chromatography Analysis

Samples (1 mL each) were centrifuged (5000 g, 10 min); the supernatant was filtered through a 0.45-µm Millipore membrane filter and then analyzed at a Kristal-5000.2 gas chromatograph (Khromatek, Podolsk, Russia) using a flame ionization detector and a BP1 J08 capillary column (60 m × 0.30 mm × 0.53 μm) (SGE Analytical Science, Ringwood, Australia). The conditions for the analysis were as follows: column thermostat temperature, 150 °C; evaporator temperature, 310 °C; detector temperature, 250 °C; carrier gas (helium) flow rate, 5.2 cm^3^/min.

The amount of CAP was determined by the calibration curve. The curve was plotted using twofold serial dilutions of an aqueous solution of CAP at an initial concentration of 2.0 mg mL^−1^.

### 2.9. Statistical Data Processing

All experiments were performed in triplicate and statistical methods such as standard deviation and standard error were employed using [22]. The confidence level at 95% intervals was calculated for each set of samples to determine the margin of error.

## 3. Results

### 3.1. Morphology and Ultrastructure of Cells

Cells of strain BS3 are immotile, rod-shaped with spherically rounded ends, and approximately 1.0–1.4 μm long and 0.6–0.8 μm wide. Cells multiply by binary division and do not form spores and prosthecas. Electron microscopy of ultrathin sections revealed the cell wall structure typical of Gram-positive bacteria. The outer surface of the cell wall is bounded by a monolayer contour of unknown nature. The cell cytoplasm has extensive electron-transparent inclusions of up to 200–300 nm in diameter (Figure 1).

After 5–6 days of incubation at 28 °C on an agarized mineral medium supplemented with caprolactam, the bacterium forms small, rounded colonies 2–3 mm in diameter, which is grayish-white, slimy, non-transparent, homogeneous; with a slightly convex profile, smooth edge, and smooth surface.

### 3.2. Physiological and Biochemical Properties

The physiological and biochemical characteristics of strain BS3 are given in Table 1. The isolate is a chemoorganotrophic, catalase-positive, and oxidase-negative organism. It possesses a respiratory type of metabolism, is capable of nitrate reduction; liquefies gelatin, hydrolyzes casein, and starch. Does not degrade esculin and tweens, and does not form indole. Tests for β-galactosidase, arginin dihydrolase, and urease are negative. As a source of carbon, the strain uses a broad range of organic compounds, including simple and complex carbohydrates, some alcohols, and organic acids.

Strain BS3 grows within the temperature range of 10 up to 37 °C and pH 6.0 up to 10.5, optimally at 25–28 °C and pH 7.5–8.0, which enables attributing it to mesophilic, non-acid-resistant, and alcalotolerant bacteria. The culture is capable of growing at the high salinity of the medium (up to 18%), though the amount of salt for optimal growth is 0.2%. These data indicate that the investigated strain belongs to halotolerant bacteria.

### 3.3. Identification of the Strain BS3

For the phylogenetic analysis of strain BS3, we determined the 16S rRNA gene sequence of 1283 bp (GenBank accession no. JN787123). Analysis of the sequence showed the isolate to belong to the class Actinobacteria, genus *Brevibacterium* and to have the highest similarity of 16S rRNA gene to type strain *B. epidermidis* VKM Ac-2108T (99.8%). The similarity between 16S rRNA genes of strain BS3 and the type strains of other most closely related species of this genus—*B. iodinum* DSM 2062T and *B. permense* VKM Ac-2280T—was lower, 98.8%. The mean DNA–DNA relatedness value between strain BS3 and *B. epidermidis* VKM Ac-2108T was 71%. The G + C content in the DNA of strain BS3 was 62.6 mol. %, which is typical of *B. epidermidis* bacteria. Based on the data obtained, strain BS3 was attributed to the species *Brevibacterium epidermidis*. The phenotypic characteristics of strain BS3 (Table 1) also confirm its phylogenetic identification as a representative of *Brevibacterium epidermidis* [23].

### 3.4. Caprolactam Tolerance

As we mentioned above, CAP is a toxic compound and at certain concentrations, it inhibits bacterial growth, including those of degrading bacteria. The rate and degree of caprolactam degradation in wastewater, first, depends on the amount of this compound, hence the tolerance of the microorganisms-destructors to high concentrations of caprolactam can be critical in practical use. To determine the caprolactam tolerance of BS3, we compared the growth parameters of the culture (duration of the lag phase, maximal specific growth rate, OD590, dry weight of cells) during its cultivation in an MSM containing from 0.1 up to 25.0 g/L substrate. The results obtained are presented in Table 2 and Figure 2.

In the experiments, BS3 was found not to grow at a substrate concentration lower than 0.5 g/L (Table 2). The lag phase at a CAP concentration of 0.5 g/L was minimal as compared with other variants of the experiment; the culture began to grow almost immediately after the inoculum was introduced (Table 2 and Figure 2). The stationary growth phase was rapidly reached, but the optical density was not high (0.81), which, most likely, indicates that the substrate was depleted, and cells ceased to grow. At a CAP concentration of 1.0 and 2.0 g/L the culture had the maximal values of optical density (1.57 and 1.50, respectively), dry weight of cells (0.98 and 0.92, respectively), and specific growth rate (0.236 and 0.224 h^−1^, respectively) (Table 2). Though the bacterial growth began after a small lag period (6 and 7 h, respectively), we believe these concentrations of the substrate to be optimal. A further increase in the amount of CAP in the medium was accompanied by a proportional increase in the lag period, a decrease in the specific growth rate, as well as a reduction of the optical density. At a substrate concentration of 10.0 g/L and higher, the value of μmax was observed to decrease dramatically; however, the culture did not cease to grow completely. A critical amount of CAP in the medium, at which bacterial growth was still observed, was 22.0 g/L. The duration of the lag phase, in that case, was more than 60 h, and the values of OD and μmax were minimal (0.15 and 0.022 h^−1^, respectively) (Table 2 and Figure 2). At higher concentrations of the xenobiotic, as well as in controls (without the addition of substrate and/or inoculum), there was no culture growth.

### 3.5. Caprolactam Degradation

Based on the results of the previous experiment, the CAP degradation dynamics by strain BS3 were investigated at an optimal concentration of substrate, 1.0 g/L. Cell growth was monitored by the change of optical density, and the content of substrate in samples was assessed by gas chromatography. As shown in Figure 3, the investigated strain totally utilized this amount of CAP over 160 h. The substrate was mainly consumed during the active growth phase (40–120 h), its content in the medium over this time decreased by 91.8%; the maximal value of optical density was 1.58. An insignificant consumption of CAP also continued in the stationary growth phase (120–160 h) to make about 5.2%. In the control experiment without adding the inoculum, the concentration of caprolactam did not change, which indicates the absence of bacterial growth (Figure 3, curve 3).

Each value is a mean of three replicates with error bars indicating the standard deviation from the mean.

### 3.6. Growth of Strain BS3 on Caprolactam Intermediates and Nylon Oligomers

It is known that caprolactam degradation starts with a caprolactam ring opening to form 6-aminohexanoic acid, followed by the deamination to 6-oxohexanoate and subsequent oxidation to yield adipic acid, which is then converted via *β*-oxidation reactions of the fatty acid metabolic pathway [10,24]. It is important to note that 6-AHA is also a product of the hydrolysis of nylon oligomers (6-AHA oligomers) [1,2,10,25]. Thus, the degradation of OLN can theoretically proceed via the CAP catabolic pathway (Figure 4).

For this reason, we studied the growth of strain BS3 in an MSM containing OLN, 6-AHA, or ADA as sole sources of carbon and energy. The concentrations of substrates were 1.0 g/L. Especially for this experiment, we synthesized the 6-AHA linear dimer and trimer. The results obtained showed that BS3 was able to grow on all substrates tested (Figure 5). This implies that the investigated strain can utilize the linear nylon oligomers as a sole source of carbon and energy. Additionally, degradation of CAP and OLN most likely occurs in BS3 via the common biochemical pathway, the formation of 6-AHA and ADA. The lack of information about genes encoding the key enzymes of CAP catabolism did not enable a strict molecular genetic proof of this assumption. It should be noted that the growth rate, as the optical density of the culture in a medium supplemented with OLN, was lower than on 6-AHA and/or ADA. Bacterial growth on OLN began after a lengthier lag phase than on other substrates, which led to an increase in the total culture growth time.

Each value is a mean of three replicates with error bars indicating the standard deviation from the mean.

## 4. Discussion

Interest in research into various aspects of microbial degradation of caprolactam and nylon-6 has considerably grown in recent years. This is because the volume of production of CAP and polymers on its basis constantly increases, and toxic production wastes enter the environment. Initially, the ability to degrade CAP was found in *Pseudomonas* bacteria, which are known to possess metabolic plasticity and can utilize a broad range of organic compounds, including toxic pollutants [26]. In the opinion of Iizuka et al. [27], this is due to a lactamase activity, which pseudomonads possess, and which rarely occurs in other microorganisms. We have shown that the ability of *Pseudomonas* bacteria to utilize CAP is determined by plasmids (CAP plasmids), which carry genetic information required for the complete mineralization of the xenobiotic [11,28]. Later works described caprolactam-utilizing bacteria of the genera *Alcaligenes*, *Corynebacterium*, *Acinetobacter*, *Achromobacter*, *Arthrobacter*, *Microbacterium*, *Bacillus,* and others [6,8,9,10,29]. A phylogenetic study of CAP degraders isolated from nylon-6 production wastes attributed them to three phyla: Proteobacteria (Beta- and Gammaproteobacteria), Actinobacteria, and Firmicutes [7]. The plasmid or chromosomal localization of CAP catabolism genes in degraders representing these taxonomic groups remains an open issue.

It is interesting that while several decades ago CAP degraders have been isolated only from soils polluted by CAP and nylon-6 production wastes, in recent years they have been also found in soils with nonspecific pollutants [30]. The emergence of the ability to degrade CAP in bacteria can be contributed to the pollution of the environment by CAP polymerization products (car and aircraft tire treads, fishing nets, cloths, etc.) and the adaptation of bacteria to them owing to the extension of enzymes’ substrate specificities, as well as the spreading of CAP plasmids in bacterial communities.

The present study focused on the characterization of gram-positive caprolactam-degrading bacterium BS3 with the view of investigating its potential for the biological treatment of caprolactam and nylon-6 production wastes. This microbe was isolated from soil samples taken in a CAP production territory (Russia). Based on the detailed investigation of its morphological, physiological, and biochemical properties, 16S rRNA gene sequencing and data of DNA–DNA hybridization with closely related type strain *B. epidermidis* VKM Ac-2108T, BS3 was identified as belonging to the phylum Actinobacteria, species *Brevibacterium epidermidis*. Though actinobacteria are known to be capable of degrading a broad range of natural and man-made organic compounds, CAP degraders of the genus *Brevibacterium* are described here for the first time. According to the literature data, *B. epidermidis* bacteria are typical representatives of human skin microbiota [31].

Strain *B. epidermidis* BS3 grew within a broad range of concentrations of the xenobiotic, from 0.5 up to 22.0 g/L. An optimal concentration of CAP was 1.0 g/L. At a concentration greater than 10.0 g/L, the CAP had a negative effect on the growth of the culture, as evidenced by an increase in the lag period and a decrease in the maximal specific growth rate and optical density of the culture. A study of CAP degradation dynamics by gas chromatography showed the bacterium to degrade completely this amount of substrate over 160 h. Its major amount, about 92%, was consumed by cells in the exponential growth phase. It should be noted that strain BS3 utilized CAP without extra sources of carbon or growth factors, in contrast to bacteria that require additional carbon sources or vitamins to degrade this xenobiotic [32].

Of special interest is that BS3 possesses a unique capability of utilizing not only CAP but also the linear nylon oligomers (dimer and trimer), which distinguishes it from the earlier described narrow-specificity CAP and/or oligomers’ degraders. As we mentioned above, the distinction between CAP and OLN biodegradation processes is the hydrolysis reaction of the amide bond in the substrate molecule, as the result of which a 6-AHA species forms in both cases; it can be further oxidized by CAP or OLN degraders via the common biochemical pathway. Despite this, the earlier described CAP degraders are incapable of utilizing OLN [10,11]. In turn, the bacteria *Alcaligenes* sp. D-2 and *Agromyces* sp. KY5R that utilize the oligomers do not degrade CAP [33,34]. A work by Baxi [2] has shown that, at the introduction of caprolactam-degrading strain *Alcaligenes faecalis* into a non-sterile soil polluted by nylon-6 production wastes, the amounts of CAP and linear oligomers in the soil are observed to decrease. However, no strict proof has been given that these pollutants are degraded by namely *A. faecalis*. The ability to degrade CAP and some linear and cyclic OLN has been described only for *Corynebacterium aurantiacum* B2, but this strain utilizes CAP and its derivatives only upon the addition of yeast extract to the medium [35]. In contrast, the isolate investigated in this work had no requirement of yeast extract/growth factors to degrade CAP and linear OLN.

One of the reasons why bacteria are incapable of utilizing simultaneously CAP and OLN is, in our mind, the narrow substrate specificity of 6-aminohexanoate-dimer hydrolase (EC 3.5.1.46), which breaks the amide bond in the molecule of linear OLN. Thus, Kinoshita et al. [36] have shown that in *Flavobacterium* sp. KI72 6-aminohexanoate-dimer hydrolase is inactive towards 100 various compounds with the amide bond, among them peptides, linear and cyclic amides, CAP including. As for caprolactam hydrolase (EC 3.5.2.-), this enzyme has not been isolated yet and its properties have not been studied. In our view, the ability of strain BS3 to utilize both CAP and OLN can be due to the occurrence of two specific hydrolases, one of which is active toward the oligomers and the other to CAP. At the same time, we cannot completely rule out the existence in this bacterium of one enzyme with a broad substrate specificity, possessing the activities of both 6-aminohexanoate-dimer hydrolase and caprolactam hydrolase. Expansion of substrate specificity in the already available bacterial enzymes is one of the mechanisms for the emergence of enzyme activities to earlier non-degradable synthetic substrates, OLN including. Thus, at a prolonged incubation of Pseudomonas aeruginosa PAO in a mineral medium with the linear dimer as a sole source of carbon, the bacterium adapted to the xenobiotic and acquired an ability to degrade it [37]. For example, Rybkina et al. [38] have described other examples of expanded substrate specificity of enzymes that enable microorganisms to attack not only the target substrate but also its derivatives, which are persistent toxicants. Thus, the bacterial strain studied in this work possesses unique enzyme activities and can utilize CAP, 6-AHA, and low molecular weight linear OLN, which makes it promising for the biological cleanup of CAP and nylon-6 production wastes containing CAP, 6-AHA, and low molecular weight oligomer fractions.

The possibility of using microorganisms for the biological cleanup of production wastes containing CAP and OLN has only started to be investigated. The biological treatment of wastewaters using activated sludge or activated sludge mixed with specific caprolactam-degrading bacteria has been reported [2,39]. Kulkarni and Kanekar have demonstrated the application of *Pseudomonas* strains for the degradation of caprolactam from nylon-6 effluents [4]. Strain *Alcaligenes faecalis* also grew in a synthetic medium containing only the components of oligomeric waste as nutrients [2]. However, these experiments were carried out in laboratory conditions. The degradation of caprolactam in CAP and nylon-6 production wastes and the in-situ bioremediation of soil polluted with CAP have not been reported.

The rate and degree of CAP degradation in wastewater have been shown to depend primarily on the concentration of the xenobiotic [40]. According to the available data, the amount of CAP can vary from 1360 up to 3600 mg/L depending on the facility, and the content of CAP in solid wastes can reach 34% (*w*/*v*) [39]. On this basis, the tolerance of microbial degraders to high concentrations of the xenobiotic can be of crucial significance for their use in biological cleanup technologies. As a rule, the concentration of CAP optimal for most microbial degraders is 1.0–2.0 g/L [8,41,42]. An increase in the concentration of substrate up to 5.0 g/L led to a significant reduction in the cell growth and harvest rates in *Pseudomonas* bacteria, and at a concentration of 10.0 g/L, they ceased to grow completely [41]. The maximum tolerance of *Achromobacter guttatus* KF71 was approximately 5.0 g caprolactam per L and required the supplementation of yeast to the medium [43]. Bacteria *Proteus* sp. NTS2 and *Achromobacter* sp. grew at 10.0 and 15.0 g/L CAP, respectively; however, these amounts of substrate were not optimal and inhibited cell growth to a significant degree [6,7]. Strain *Acinetobacter calcoaceticus* demonstrated poor growth at 19.0 g/L CAP [8]. A critical amount of CAP in a mineral medium, at which the investigated strain continued to grow, was 22.0 g/L, which is higher than in earlier reported caprolactam-degrading bacteria. These results demonstrate that strain BS3 has a great potential for bioremediation of caprolactam from production wastes within a broad range of concentrations of the pollutant.

It is generally recognized that efficient uptake of organic pollutants from water and/or soil by microorganisms is impeded by such factors as pH values, temperature as well as salinity non-optimal for their growth. Therefore, not all microbial degraders are suitable for environmental cleanup technologies. Since wastes from CAP and CAP-based polymer manufacture are alkaline solutions [44], halo- and alcalohalotolerance of strain *B. epidermidis* BS3—along with its ability to utilize a broad range of toxic pollutants and high CAP tolerance—make it promising for use in technologies of the biological cleanup of CAP and nylon-6 production wastes under conditions distinct from the optimum.

## Figures and Tables

**Figure 1 microorganisms-11-00373-f001:**
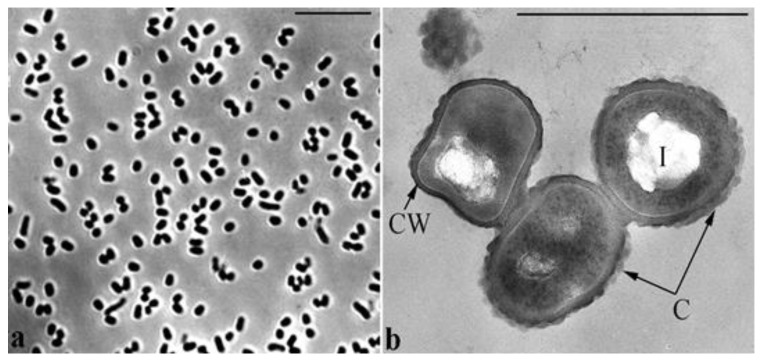
Morphology of strain BS3 cells: (**a**) phase contrast; scale bar, 10 μm; (**b**) ultrathin section; scale bar, 1 μm. Designations: CW, cell wall; C, capsule; I, electron-transparent inclusion.

**Figure 2 microorganisms-11-00373-f002:**
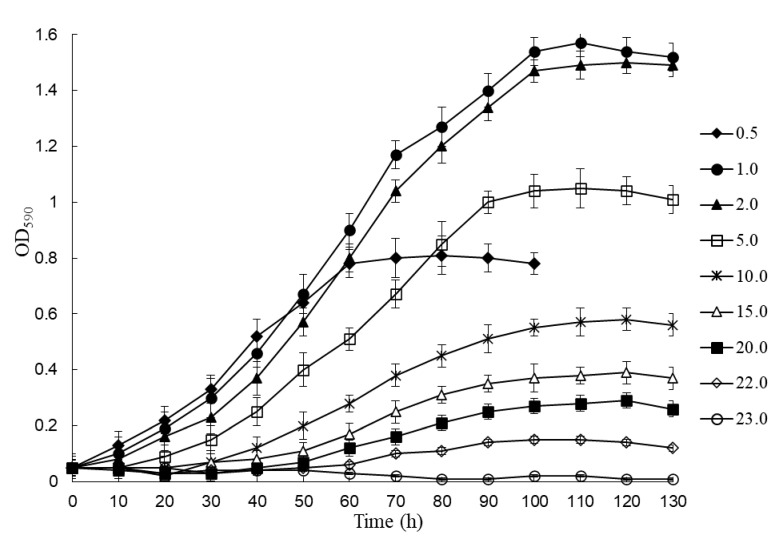
Growth dynamics of *Brevibacterium epidermidis* BS3 in a mineral medium at different concentrations of caprolactam (g/L) as a sole carbon and energy source. Each value is a mean of three replicates with error bars indicating the standard deviation from the mean.

**Figure 3 microorganisms-11-00373-f003:**
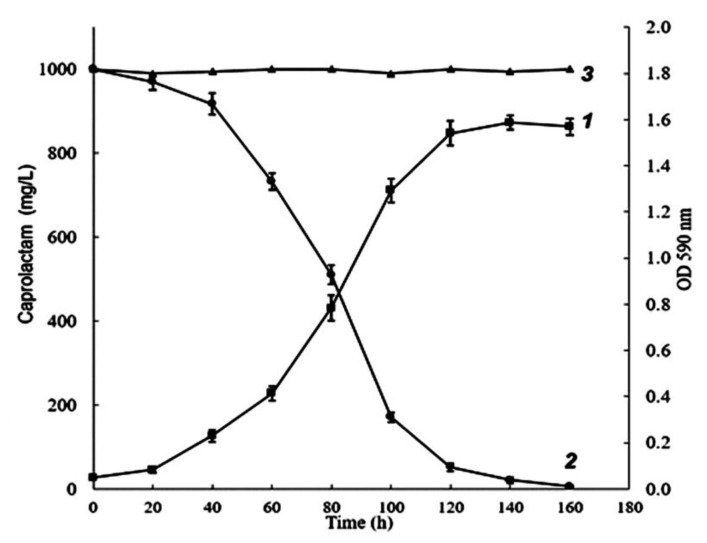
Growth (1) and utilization dynamics (2,3) of *Brevibacterium epidermidis* BS3 in a mineral medium supplemented with caprolactam (1.0 g/L) as a sole carbon and energy source. 3—the control variant without adding the inoculum.

**Figure 4 microorganisms-11-00373-f004:**
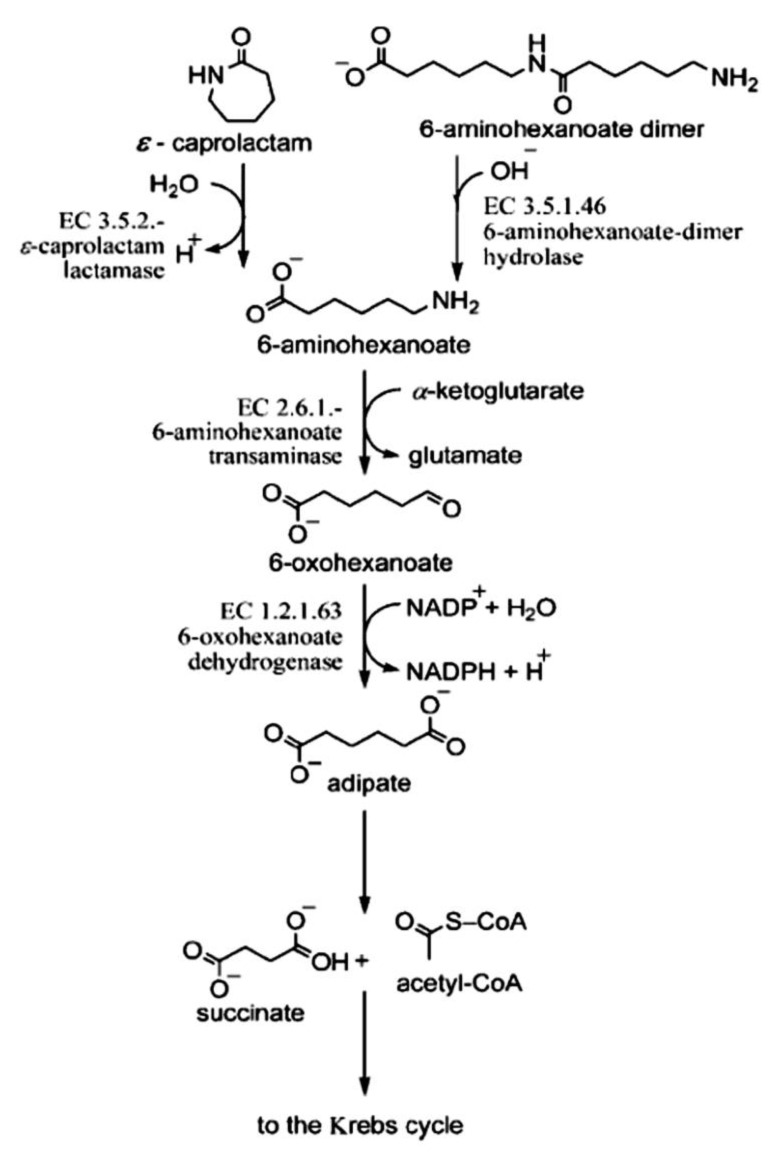
Scheme for the biodegradation of caprolactam and linear nylon oligomers (as exemplified by the dimer).

**Figure 5 microorganisms-11-00373-f005:**
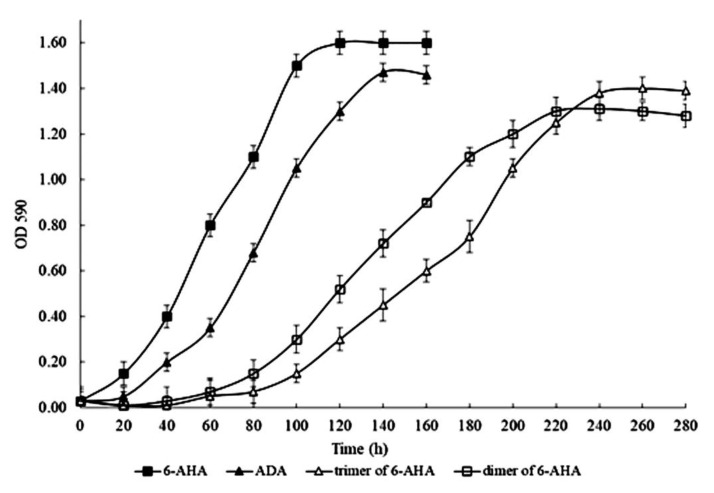
Growth dynamics of *Brevibacterium epidermidis* BS3 in a mineral medium supplemented with linear nylon oligomers and caprolactam intermediates as sole sources of carbon and energy (concentration of substrates, 1.0 g/L).

**Table 1 microorganisms-11-00373-t001:** Morphological, physiological, and biochemical characteristics of strain BS3.

Characteristics	
Cell form/size	Round-ended short rods, 1.0–1.4 × 0.6–0.8 μm
Gram stain	Positive
Motility	Negative
Relation to oxygen	Aerobic
Oxidase	Negative
Catalase	Positive
Nitrate reduction	Positive
Urease	Negative
Esculin hydrolysis	Negative
Starch hydrolysis	Negative
Casein hydrolysis	Positive
Gelatin liquefaction	Positive
*β*-Galactosidase, arginine dihydrolase	Negative
Indole production	Negative
Hydrolysis of Tweens	Negative
Growth temperature (optimum), °C	10–37 (25–28)
Growth pH range (optimum)	6.0–10.5 (7.5–8.0)
NaCl tolerance, % (optimum)	0–18.0 (0.2)
DNA G + C content, mol. %	62.6
Assimilation of substrates	
Fructose, cellobiose, galactose, glucose, ribose, sucrose	Positive
Xylose, arabinose, maltose, rhamnose, fucose, lactose	Negative
Dulcitol, inositol, adonitol, arobitol, sorbitol	Negative
Mannitol, glycerol	Positive
Malate, phenylacetate, sodium citrate, potassium gluconate, capric acid, succinate	Positive
Formation of acid from	
Inositol	Negative
Mannitol	Positive
Rhamnose	Negative
Salicylate	Negative
Sorbitol	Negative

**Table 2 microorganisms-11-00373-t002:** Growth values of *Brevibacterium epidermidis* BS3 vs. caprolactam concentrations in a mineral medium.

CAP Concentration, g/L	Lag Phase, h	µ_max_, h^−1^	OD_590_	Dry Cell Weight, g/mL
0.4	0	0	0.03	0
0.5	1.0	0.143	0.81	0.48 ± 0.03
1.0	6.0	0.236	1.57	0.98 ± 0.09
2.0	7.0	0.224	1.50	0.92± 0.07
5.0	15	0.146	1.05	0.64 ± 0.05
10.0	27	0.080	0.58	0.31 ± 0.02
15.0	40	0.0543	0.39	0.24 ± 0.02
20.0	50	0.041	0.29	0.176 ± 0.01
22.0	60	0.022	0.15	0.08 ± 0.001
23.0		0	0.03	0

The table shows the mean values ± standard deviations. The results were obtained from three independent experiments. The maximum values of optical density at the beginning of the stationary phase of growth are given. The weight of dry cells was determined at the point of maximum optical density according to the calibration curve.

## Data Availability

The data presented in this study are available on request from the corresponding author.

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
