# Peer review of "Epsilon-Caprolactam- and Nylon Oligomer-Degrading Bacterium Brevibacterium epidermidis BS3: Characterization and Potential Use in Bioremediation"

_microorganisms, 2023, doi:10.3390/microorganisms11020373_

Round 1

Reviewer 1 Report

In this study, a caprolactam-degrading bacterial strain, Brevibacterium epidermidis BS3, was isolated and identified from polluted soils. Strain BS3 is an alcalo- and halotolerant organism, grows within a broad range of CAP concentrations, and degrades 1.0 g/L CAP over 160 h. Furthermore, strain BS3 is also capable of utilizing linear nylon oligomers, CAP polymerization by-products, as sole sources of carbon and energy. The broad range of utilized toxic pollutants, the tolerance for high CAP concentrations, as well as the physiological properties of BS3 determine the prospects of its use for the biological cleanup of CAP and nylon-6 production wastes.

The manuscript is easy to understand, the illustrations, tables and figures in this manuscript are useful and necessary. The objective is clear and the references are mostly relevant. However, there are still some revisions needed, listed as following:

1. Figure 2. Data points are missing error bars. Is there no duplication? In addition, there's a little problem with the numerical format in the vertical coordinates. It should be a decimal point, not a comma.

2. Figure 3. Similarly, the numerical format in the right vertical coordinates needs to be revised, it should be a decimal point, not a comma. The value or unit of the left vertical axis is incorrect. The maximum value is 1000 g/L?

3. Figure 5. It is suggested that the stable period of bacterial growth curve should be detected.

4. After growing with linear nylon oligomers or caprolactam as the sole sources of carbon, the intermediates produced by strain BS3 should be detected and identified.

Author Response

Cover letter

We are grateful to the reviewers for their careful reading of our article and for their valuable comments. Below are the responses to the comments of the reviewers. Appropriate corrections have been made to the text of the article.

Responses to the comments of the reviewer 1.

  1. Figure 2. Data points are missing error bars. Is there no duplication? In addition, there's a little problem with the numerical format in the vertical coordinates. It should be a decimal point, not a comma.

Experiments were performed in triplicate and statistical methods for data processing were applied. Initially, to improve the perception of the graphs, the standard deviations of the mean are not indicated. At the request of the reviewer, error bars indicating the standard deviation from the mean were added. Also, the format of numbers on the vertical coordinate axis has been corrected.

  1. Figure 3. Similarly, the numerical format in the right vertical coordinates needs to be revised, it should be a decimal point, not a comma. The value or unit of the left vertical axis is incorrect. The maximum value is 1000 g/L?

The format of numbers in the vertical coordinate axis has been corrected. Caprolactam concentration value has also been corrected.

  1. Figure 5. It is suggested that the stable period of bacterial growth curve should be detected.

Changes have been made in accordance with the comments of the reviewer. The stable period of bacterial growth has been shown based on experimental data obtained in our study.

  1. After growing with linear nylon oligomers or caprolactam as the sole sources of carbon, the intermediates produced by strain BS3 should be detected and identified.

Currently, in related literature, only one biochemical pathway for degradation of caprolactam in bacteria via generation of 6-aminohexanoic acid and adipic acid has been described. In the 60s of the 20th century, Japanese and Russian scientists predicted this pathway after identification of metabolites in culture liquid.

Naumova, R.P.; Belov, I.S. Conversion of ε-fminocaproic acid by bacterial decomposition of caprolactam. Biokhimya (Article in Russian) 1968, 33, 946-952.

Naumova, R.P. Bacterial degradation of ε-caprolactam. Prikl. Biochim. Mikrobiol. (Article in Russian) 1978, 5, 43-46.

Naumova, R.P.; Esikova, T.Z.; Ilyinskaya, O.N.; Grishchenkov, V.G.; Boronin, A.M. Metabolism of e-caprolactam in Pseudomonades containing plasmids of its degradation. Mikrobiologia (Article in Russian) 1988, 57, 426–430.

Kinoshita, S.; Kobayashi, E.; Okada, H. Degradation of e-caprolactam by Achromobacter guttatus KF71. J. Ferment. Technol. 1973, 51, 719–725.

Fukumura, T. Bacterial breakdown of ε-caprolactam and its cyclic oligomers. Plant Cell Physiol. 1966, 7(1), 93–104.

Also, it has been shown that degradation of oligomers leads to the formation of 6-aminohexanoic acid that is further metabolized by the route of caprolactam degradation.

Shama, G.; Wase, D.A. The biodegradation of ε-caprolactam and some related compounds. Int. Biodeterior. Bull. 1981, 17(1), 1–7

Fukumura, T. Bacterial breakdown of ε-caprolactam and its cyclic oligomers. Plant Cell Physiol. 1966, 7(1), 93–104.

Negoro, S. Biodegradation of nylon and other synthetic polyamides. Biopolymers. 2002, 9, 395–415.

Some citations are given in the article.

This pathway is considered proven and is not doubted by scientists who work in this field of study. Scheme of catabolism is shown in all publications devoted to different aspects for the degradation of caprolactam (CAP) and nylon oligomers (OLN). But none of these articles has reported that the scientists again have carried out research on metabolite identification in culture liquid that is intrinsically difficult to perform. Some metabolites such as 6-oxohexanoate and 6-aminohexanoate are rapidly oxidized and it is difficult to detect them. We agree that the growth of strain BS3 on the intermediates of CAP is an indirect conformation that CAP and OLN degradation occurs via the said biochemical pathway. It goes without saying that direct and easier (compared to identification in culture liquid) evidence could be identification of genes of CAP catabolism in the BS3 strain. However, key genes are still not identified, so, there is no relevant information in databases. In the light of it, it seems impossible to identify genes of CAP catabolism in strains-destructors.

In addition, the main goal of our present study was to investigate the morphological, physiological, biochemical properties and the taxonomic status of the new strain BS3, and assess prospects of its application for the biological cleanup of industrial wastes from production of CAP and CAP-based polymers. The title of the article reflects the main idea of our study.

The authors once again express their gratitude to the referee for the formulated remarks. Corrections made in accordance with them have improved the article. On behalf of all co-authors, Inna Solyanikova

Reviewer 2 Report

Numerous studies have described the degradative capacity of different microorganisms in relation to xenobiotic compounds. New contributions in this regard must not only prove the ability of the microorganism to carry out such action, but also provide information on specific metabolic and genetic aspects. In the present case, this premise is not fulfilled, since only a preliminary study of the degradative capacity of the isolate under laboratory conditions is presented. It would be necessary to include additional assays to complete the data presented in one way or another.

Minor suggestions:

1. Italicize all scientific names.

2. The standard deviation data should be included in Figure 2. If this makes it difficult to visualize the curves correctly, an alternative presentation format (Table) should be considered.

3. The legend of Table 2 is not sufficiently self-explanatory. To what time do the OD and cell dry weight data refer?

4. Citation 26 does not appear in the text.

5. In L. 348 "phylum Actinobacteria" should be deleted, since it is repeated.

6. Change the term microflora to microbiota.

7. What does the second part of the sentence "B. epidermidis bacteria are typical representatives of human skin microflora and are found the most often in clinical specimens

Author Response

Cover letter

We are grateful to the reviewers for their careful reading of our article and for their valuable comments. Below are the responses to the comments of the reviewers. Appropriate corrections have been made to the text of the article.

Responses to the comments of the Reviewer 2.

Numerous studies have described the degradative capacity of different microorganisms in relation to xenobiotic compounds. New contributions in this regard must not only prove the ability of the microorganism to carry out such action, but also provide information on specific metabolic and genetic aspects. In the present case, this premise is not fulfilled, since only a preliminary study of the degradative capacity of the isolate under laboratory conditions is presented. It would be necessary to include additional assays to complete the data presented in one way or another.

A starting point to study metabolism in the mid-20th century was the need to develop a biological approach for treatment of wastewater during caprolactam and nylon production. Since that time, a significant amount of research has been devoted to the study of biochemical and genetic aspects of the degradation of caprolactam(CAP) and nylon oligomers (OLN). Some of these reports are as follows:

Fukumura, T. Bacterial breakdown of ε-caprolactam and its cyclic oligomers. Plant Cell Physiol. 1966, 7(1), 93–104.

Shama, G.; Wase, D.A. The biodegradation of ε-caprolactam and some related compounds. Int. Biodeterior. Bull. 1981, 17(1), 1–7

Boronin, A.M.; Naumova, R.P.; Grishchenkov, V.G.; Ilijinskaya, O.N. Plasmids specifying ε-caprolactam degradation in Pseudomonas strains. FEMS Microbiol. Lett. 1984, 22, 167–170.

Esikova, T.Z.; Grishchenkov, V.G.; Boronin, A.M. Plasmids that control ε-caprolactam biodegradation. Microbiol. (Moscow). 1990, 59, 547–552.

Negoro, S. Biodegradation of nylon and other synthetic polyamides. Biopolymers. 2002, 9, 395–415.

Yasura, K.; Uedo, Y.; Takeo, M.; Kato, D.; Negoro, S. Genetic organization of nylon-oligomer-degrading enzymes from alcalophilic bacterium, Agromyces sp. KY5R. J. Biosci. Bioeng. 2007, 104, 521–524. doi: 10.1263/jbb.104.521

Esikova, T.Z.; Ponamoreva, O.N.; Baskunov, B.P.; Taran, S.A.; Boronin, A.M. Transformation of low-molecular linear caprolactam oligomers by caprolactam-degrading bacteria. J. Chem. Technol. Biotechnol. 2012, 87, 1284–1290. DOI 10.1002/jctb.3789

Otzen, M.; Palacio, C.; Janssen, D.B. Characterization of the caprolactam degradation pathway in Pseudomonas jessenii using mass spectrometry-based proteomics. Appl. Microbiol. Biotechnol. 2018, 102, 6699–6711.

Scheme for xenobiotic catabolism has been proposed by Japanese and Russian scientists in the 60s of the 20th century. It was formulated based on the study of metabolites in culture liquid of the strains-destructors. Citations are given in the text of the article. This pathway is the only pathway that was described in related literature and is not doubted. Hence, intrinsically difficult identification of CAP metabolites in culture liquid of bacteria able to degrade CAP and OLN is not usually retried. Genetic aspects of CAP degradation are still poorly understood. Key genes of CAP (and consequently of OLN) catabolism are not identified. That is why it turns out impossible to identify genes of CAP catabolism in the studied strain.

This article focuses on characterization of the strain-destructor Brevibacterium epidermidis BS3. The main object was to study morphology, physiological, biochemical properties and taxonomic status of the new strain. This is the first to report the ability of Brevibacterium to utilize CAP. This description contributes to further gain comprehensive knowledge about the ability of Gram-positive bacteria to utilize this xenobiotic. The aim of this work was also assess prospects of application of BS3 for the biological cleanup of industrial wastes from production of CAP and CAP-based polymers. The title of the article reflects the main idea of our study. It has been shown that unlike caprolactam- and oligomer-degrading strains previously described, the new strain has a unique ability to utilize both CAP and OLN. The ability to degrade a wide range of synthetic substrates makes this promising bacterium a potent candidate for in situ bioremediation of industrial wastes from production of CAP and nylon containing CAP and fractions of low molecular OLN.

  1. Italicize all scientific names.

All designations of bacteria in the article are italicized. As to Actinobacteria, Proteobacteria (Beta- and Gammaproteobacteria) and Firmicutes, to the best of our knowledge, the names of bacterial classes and phyla are not italicized. Please, indicate the line in the text of the article where the scientific name should be italicized.

  1. The standard deviation data should be included in Figure 2. If this makes it difficult to visualize the curves correctly, an alternative presentation format (Table) should be considered.

The graph has been corrected in accordance with the comments of the reviewer

  1. The legend of Table 2 is not sufficiently self-explanatory. To what time do the OD and cell dry weight data refer?

Thank you for your comment. The corresponding amendments are made to the legend of Table 2. Now the legend reads: the maximum values of optical density at the beginning of the stationary phase of growth are given. The weight of dry biomass was determined at the point of maximum optical density according to the calibration curve

  1. Citation 26 does not appear in the text.

corrections were made

  1. In L. 348 "phylum Actinobacteria" should be deleted, since it is repeated.

corrections made

  1. Change the term microflora to microbiota.

corrections made

  1. What does the second part of the sentence "B. epidermidis bacteria are typical representatives of human skin microflora and are found the most often in clinical specimens?

Phrase is corrected. In this sentence we‘d like to underline that the studied strain-destructor was identified as B. epidermidis, representatives of which usually inhabit human skin and are often isolated from biological materials such as blood, serum, plasma, faeces, and tissues, collected from humans in clinical settings. Since the second part of this sentence raises the question, we decided to delete it.

The authors once again express their gratitude to the referee for the formulated remarks. Corrections made in accordance with them made it possible to improve the article.

The authors once again express their gratitude to the referee for the formulated remarks. Corrections made in accordance with them have improved the article.

On behalf of all co-authors,

Inna Solyanikova

Round 2

Reviewer 1 Report

All comments have been addressed.

Reviewer 2 Report

The authors argue in their answers the validity of works such as the one presented by them. Although I believe that it is feasible to carry out tests that deepen to a greater extent the knowledge of the described degradative process, they have provided reasons to consider this study as publishable, in its current state. All other suggestions have been taken into consideration.